# Extracellular RNAs as Biomarkers of Sporadic Amyotrophic Lateral Sclerosis and Other Neurodegenerative Diseases

**DOI:** 10.3390/ijms20133148

**Published:** 2019-06-27

**Authors:** Takashi Hosaka, Takenari Yamashita, Akira Tamaoka, Shin Kwak

**Affiliations:** 1Department of Neurology, Division of Clinical Medicine, Faculty of Medicine, University of Tsukuba, Tsukuba, Ibaraki 305-8575, Japan; 2Department of Pathophysiology, Tokyo Medical University, Shinjuku-ku, Tokyo 160-8402, Japan; 3Department of Molecular Neuropathogenesis, Tokyo Medical University, Shinjuku-ku, Tokyo 160-8402, Japan

**Keywords:** neurodegenerative disease, amyotrophic lateral sclerosis (ALS), RNA editing, adenosine deaminase acting on RNA 2 (ADAR2), extracellular RNA, biomarker

## Abstract

Recent progress in the research for underlying mechanisms in neurodegenerative diseases, including Alzheimer disease (AD), Parkinson disease (PD), and amyotrophic lateral sclerosis (ALS) has led to the development of potentially effective treatment, and hence increased the need for useful biomarkers that may enable early diagnosis and therapeutic monitoring. The deposition of abnormal proteins is a pathological hallmark of neurodegenerative diseases, including β-amyloid in AD, α-synuclein in PD, and the transactive response DNA/RNA binding protein of 43kDa (TDP-43) in ALS. Furthermore, progression of the disease process accompanies the spreading of abnormal proteins. Extracellular proteins and RNAs, including mRNA, micro RNA, and circular RNA, which are present as a composite of exosomes or other forms, play a role in cell–cell communication, and the role of extracellular molecules in the cell-to-cell spreading of pathological processes in neurodegenerative diseases is now in the spotlight. Therefore, extracellular proteins and RNAs are considered potential biomarkers of neurodegenerative diseases, in particular ALS, in which RNA dysregulation has been shown to be involved in the pathogenesis. Here, we review extracellular proteins and RNAs that have been scrutinized as potential biomarkers of neurodegenerative diseases, and discuss the possibility of extracellular RNAs as diagnostic and therapeutic monitoring biomarkers of sporadic ALS.

## 1. Introduction

Neurodegenerative diseases, including Alzheimer disease (AD), Parkinson disease (PD), and amyotrophic lateral sclerosis (ALS) are characterized by the progressive loss of specific functions in the central nervous system (CNS), resulting from structural changes in selective neuronal circuits. The observation of abnormal aggregates of proteins and/or RNAs such as amyloid, tau, and the transactive response DNA/RNA binding protein of 43kDa (TDP-43) in these neuronal circuits is a pathological hallmark of these diseases [1,2,3]. With an advance in our understanding of disease pathogenesis, an increasing number of clinical trials of potential treatments based on plausible pathogenetic hypotheses have been conducted. Once effective treatments are a reality for neurological diseases, their success will depend upon reliable diagnostic and/or therapeutic monitoring biomarkers that enable early diagnosis and the evaluation of therapeutic efficacy.

A biomarker is characterized as an indicator of normal biological processes, pathogenic processes, or pharmacologic responses to a therapeutic intervention, which is objectively measured and evaluated [4]. Good biomarkers are useful for accurate diagnosis (diagnostic biomarker), the prediction of prognosis (prognostic biomarker) and response to a certain type of treatment (predictive biomarker), the determination of dosage (pharmacodynamic biomarker), and/or disease activity (disease progression biomarker) [5]. Although the definite diagnosis of neurodegenerative diseases comes from histological observation of the spinal cords and brains, biomarkers that reflect disease-specific pathological events in the CNS should be obtained in a non-invasive or low-invasive manner, including imaging and easily accessible body fluids or tissues. Liquid biopsy is a form of biopsy targeting the patient’s body fluids, including serum and cerebrospinal fluid (CSF) and can be an alternative, less invasive method for conventional tissue biopsy [6]. Recently, cell–cell communication via extracellular vesicles, such as exosomes and microvesicles, is considered to be necessary in order to maintain tissue homeostasis [7]. Extracellular vesicles are secreted from several cell types in CNS tissue, including neurons, astrocytes, and microglia. Neuronal interactions via extracellular vesicles regulate neurite growth, synapse function, and neuronal regeneration [8,9]. Therefore, extracellular proteins and RNAs have been considered as promising biomarker candidates, of various types, for neurodegenerative diseases.

ALS is a fatal adult-onset motor neuron disease that is characterized by the selective degeneration of both upper and lower motor neurons. Progressive muscle weakness and atrophy result in respiratory failure, leading to death within a few years following disease onset [10,11]. The estimated median prevalence of ALS is 5.4 cases per 100,000 [12]. Aging is one of the strongest risk factors of ALS with a peak incidence between 60–70 years [12,13]. Although more than 50 ALS-associated genes have been identified, including superoxide dismutase 1 (*SOD-1*), fused in sarcoma (*FUS*), and chromosome 9 open reading frame 72 (*C9ORF72*) [14,15], only a small proportion of sporadic ALS patients (accounting for more than 90% of all ALS cases) carry any mutations in these genes. Although a large number of pathogenic hypotheses have been proposed, including protein misfolding, excitotoxicity, and oxidative stress [14], the mechanisms underlying motor neuron death in ALS remain elusive, even in familial ALS, and neither disease-modifying treatments nor reliable biomarkers of ALS are currently available [10].

In sporadic ALS, a pathological diagnostic hallmark is the disruption of the nuclear localization of TDP-43 (a nuclear protein involved in regulating RNA processing) and the deposition of cytoplasmic aggregates comprising misfolded abnormal variants of this protein in lower motor neurons [16,17]. However, no biomarkers reflecting the abnormal subcellular localization of TDP-43 in the lower motor neurons of patients are available for clinical use. The El Escorial criteria [18] and Awaji criteria [19] are the most widely used diagnostic criteria for ALS, and are based on the combination of clinical assessment and electrophysiological examination, in order to exclude patients with ALS-like symptoms and signs (such as cervical spondylosis) from clinical trials [20,21]. In agreement with reports that ALS patients do not exhibit weakness or skeletal muscle wasting until more than 30% of anterior horn neurons degenerate [22], and that patients are diagnosed with ALS after an average of one year from disease onset [12,21,23], we neurologists frequently observe patients who do not meet these criteria at the initial visit, but turn out to fulfill the ALS criteria after 6–12 months in the outpatient clinic. Therefore, with the recent development of mechanism-based optimistic clinical trials, the development of diagnostic biomarkers for early phase ALS and predictive and pharmacodynamic biomarkers is crucial in order to enable the initiation of treatment, before the number of motor neurons decreases to below a level that is required to effectively control skeletal muscles.

In this review, we summarize extracellular proteins and RNAs as various types of candidate biomarkers of neurodegenerative diseases—in particular sporadic ALS—and discuss the need for the development of biomarkers based on molecular pathogenesis. Moreover, we also discuss the possibility of extracellular RNAs as biomarker candidates based on excitotoxicity, which is a plausible pathogenesis of sporadic ALS.

## 2. Extracellular RNAs

RNAs, including mRNA, micro RNA (miRNA) and circular RNA (circRNA), are secreted from cells in forms such as exosomes, microvesicles, and RNA binding proteins (RBP), and are referred to as extracellular RNAs (Figure 1) [24]. These extracellular RNAs may represent the pathological process in the host cells and remain in body fluids such as blood, urine, and CSF [25], suggesting that these RNAs are potentially good disease biomarkers [24,26,27]. Recently, evidence suggesting that RNAs and RNA-binding proteins play important roles in the pathogenesis of sporadic ALS, has been increasing [15,28]. In accordance with a recent focus on RNA-mediated cell–cell communication, extracellular RNAs are regarded as hopeful new biomarker candidates of neurodegenerative diseases, including ALS.

Cells secrete cellular composite molecules into extracellular space as a package surrounded by membranes called vesicles, which are classified into four types mainly according to their size and biogenesis; exosomes, microvesicles, apoptotic bodies, and RBPs [24]. Exosomes, measuring 40–100 nm in diameter, arise from intraluminal vesicles in multivesicular bodies (MVBs) and are secreted by the fusion of MVBs with the cell membrane [7,29]. Although exosome secretion was initially considered as a way of eliminating unnecessary compounds from cells, secreted molecules are now considered intracellular communication mediators, since cell–cell communications through exosomes has been reported [30]. Microvesicles measuring 100–1000 nm in diameter arise from the budding of the cell membrane, and have also been considered to play a role in cell–cell communication due to the transport of transmembrane proteins between microglia and astrocytes, through microvesicles, activating membrane proteins [31]. Apoptotic bodies, measuring 500–2000 nm in diameter, are released through blebbing of the cell membrane of apoptotic cells, and include nucleotides and cell organelles [32]. RBPs, including high-density lipoproteins, argonaute 2, and nucleophosmin 1, also bind extracellular RNAs [33,34,35]. These extracellular vesicles and RBPs protect extracellular RNAs from digestion by ribonucleases, enabling RNAs to remain in the extracellular space [24]. Therefore, extracellular RNAs are considered to be a candidate biomarker of human diseases, including neurodegenerative diseases.

The secretion of extracellular RNAs is stimulated in some adverse conditions, including hypoxia and events activating immune responses [36], and the levels of some extracellular RNAs are changed with aging [37]. It was reported that miRNAs in exosomes, and mRNA and some circRNAs in other forms, reflect their intracellular expression profiles [38], suggesting that the analysis of extracellular RNAs may indicate the health level of cells and hence, extracellular RNAs in appropriate forms can be potential biomarkers of human diseases.

MiRNAs are short non-coding RNAs (21–24 nucleotides), and are produced by two RNase III proteins, Drosha and Dicer [39]. They regulate gene expression by binding to the 3′-untranslated regions of mRNA, and play important roles in brain development [1] and various cellular processes, including apoptosis and response to immune stimuli and stress [40]. The adenosine-to-inosine (A-to-I) conversion of RNA (RNA editing) in primary miRNA influences miRNA biogenesis and function by switching target mRNAs [41]. Changes in the expression levels of miRNAs have been detected in neurodegenerative diseases such as AD, PD, and ALS, and miRNAs regulate protein levels involved in the pathogenesis of neurodegenerative diseases, through the regulation of their host gene expression [42], suggesting that miRNAs are candidate biomarkers of neurodegenerative diseases, including ALS.

CircRNAs are produced by back-splicing, which covalently links the 3′ end of an exon to the 5′ end of an upstream exon [43], and are also detected in extracellular fluid [44]. The complementary sequence across flanking introns—and nucleotide changes due to RNA editing by adenosine deaminase acting on RNA 1 (ADAR1) in the flanking introns—influence the biogenesis of circRNA [45,46]. Intriguingly, FUS, a mutation that is a causative for familial ALS, has also been shown to be involved in the biogenesis of circRNA [47]. The functions of circRNAs are largely unknown, except for that of regulating gene expression, the sponging of miRNA, RNA splicing, and cell cycle regulation, by interacting with proteins [43]. CircRNAs were widely expressed throughout the organs, especially in the CNS, and the extracellular levels of some circRNAs are more than 10-fold higher than intracellular levels, suggesting that circRNAs are actively secreted from cells [45,48,49]. The expression levels of circRNAs are increased with cell differentiation in SH-SY5Y cells and with aging in the adult mouse brain [50,51]. The characteristic circular structure protects circRNAs from digestion by ribonucleases and provides a longer half-life compared with linear RNAs [48]. Although no circRNAs have been established as biomarkers of neurodegenerative diseases, a long half-life and an abundance in the CNS of the aged, are advantages that allow circRNAs to be candidate biomarkers in aging associated neurodegenerative diseases.

## 3. Liquid Biopsy in Neurodegenerative Diseases

Increasing numbers of clinical trials of candidate drugs that target potential causative molecules have been conducted in AD and PD, which are the first and second most prevalent neurodegenerative diseases, respectively [52,53]. Currently, molecular imaging such as amyloid positron emission tomography (PET), dopamine transporter single-photon emission computed tomography (DAT-SPECT), and ^123^I meta-iodobenzylguanidine (MIBG) myocardial scintigraphy are used as diagnostic biomarkers in clinical trials. However, high costs and a limited number of facilities where molecular imaging is available necessitate the development of more feasible biomarkers. As the examination of body fluids is widely performed, extracellular proteins and RNAs in the body fluids can be potential candidates for biomarkers, provided that proteins or RNAs represent the pathological neurodegenerative processes. In this section, we discuss the candidate diagnostic biomarkers of AD and PD investigated to date, which can be obtained by liquid biopsy.

### 3.1. Alzheimer Disease

AD is characterized by progressive memory loss, the neuropathological presence of neuronal hyperphosphorylated tau, and widespread Aβ deposition in the cortices [52]. Since Aβ peptides secreted from cells in exosomes accumulate in amyloid plaques [54], cell–cell communication via extracellular vesicles is considered to be involved in the pathogenesis of AD.

The 42 amino acid form of Aβ (Aβ42), total tau (T-tau), and phosphorylated tau (P-tau) have been identified in patients’ CSF [55]. In the CSF of AD patients, Aβ42 levels are significantly decreased compared with normal control subjects. Patients also show pathological cortical amyloid changes, and T-tau and P-tau levels are significantly increased, which is in parallel with the accumulation of neocortical neurofibrillary tangles [56]. Since the changes are highly sensitive and specific (80–90%) to AD [57], these three core biomarkers in the CSF are disease-specific diagnostic biomarkers. However, in the plasma of AD patients, changes in Aβ42 levels are opposite to the levels in the CSF and significantly increased [58,59], whereas T-tau and P-tau levels are significantly increased in concordance with the levels in CSF [60,61].

Other promising diagnostic biomarker candidates in the CSF are neurofilament light chain (NEFL), neuron-specific enolase (NSE), visinin-like protein 1 (VILIP-1), heart fatty acid binding protein (H-FABP), and chitinase-3-like protein 1 (YKL-40) [55]. NEFL is a component of the axonal cytoskeleton and is involved in the maintenance of axonal structure and function [62]. Levels in the CSF, serum, and plasma are significantly increased in AD patients [63,64,65] and in multiple sclerosis (MS) and PD as well [66]. NSE is a neuronal glycolytic enzyme that is localized in the cytoplasm, and is involved in neuronal differentiation [67]. NSE levels in the CSF are significantly increased in some [68,69,70], but not in most AD patients [71,72,73], or patients with stroke, cerebral hemorrhage, or brain trauma as well [70]. VILIP-1, a neuronal calcium sensor protein involved in calcium-mediated neurotoxicity, and H-FABP, a fatty acid binding protein involved in the uptake, transport, and metabolism of fatty acid, are potential markers of brain injury [72,74], and their levels in CSF are significantly increased [72,74,75,76,77,78]. YKL-40 is a marker of neuroinflammation, and its levels in CSF are significantly increased in some [78,79,80]—but not all—AD patients [81] and in MS patients as well [82]. Although currently none of them has been established as a reliable biomarker, recent advances in the technology enabling the detection of tiny and potential disease-specific changes in the molecules in the patient’s body fluids [59] will enable the development of these in the future.

RNAs, especially non-coding RNAs, including miRNA, long non-coding RNA (lncRNA), and circRNA, play roles in AD pathogenesis by regulating the disease onset [40], rendering extracellular RNAs candidates biomarkers of AD, such as extracellular proteins. Multiple lines of study report significant differences in the levels of miRNAs in the CSF or serum between patients with AD and control subjects [83,84,85,86,87]; the upregulation of let-7d (in plasma and whole blood), let-7f (in serum), let-7g (plasm), miR-106a (in serum), miR-106b (in serum), miR-142 (in whole blood), miR-15b (in plasma), miR-191 (in plasma), miR-3065 (in serum), miR-34a (in plasma), miR-483 (in plasma), miR-9 (in serum), miR-127 (in serum), and miR-30c (in CSF) and downregulation of let-7d (in serum), let-7g (in serum), miR-106a (in whole blood), miR-106b (in whole blood), miR-107 (in plasma and whole blood), miR-125b (in serum and plasma), miR-146a (in serum and plasma), miR-15b (in serum), miR-181c (in serum), miR-191 (in serum), miR-26a (in serum), miR-29b (in plasma and peripheral blood mononuclear cells (PBMCs)), miR-3065 (in plasma), miR-342 (in serum and plasma), miR-483 (in serum), miR-9 (in whole blood and CSF), miR-127 (in CSF), miR-184 (in CSF), and miR-195 (in CSF) have been reported. However, inconsistency among results has failed to establish diagnostic biomarkers of AD.

### 3.2. Parkinson Disease

Parkinson disease is characterized by the death of dopaminergic neurons in the substantia nigra, which leads to movement dysfunction, including resting tremor, bradykinesia, muscular rigidity, and postural and gait impairment [53]. Since motor symptoms commence at the time when as many as 50% of dopaminergic neurons have already been lost, and clinical manifestations in early stages are sometimes difficult to distinguish PD from other diseases with parkinsonism, such as progressive supranuclear palsy (PSP) and multiple system atrophy (MSA) [88,89], the development of early diagnostic biomarkers is required.

One of the most extensively investigated biomarker candidates is α-synuclein: a composite molecule of the cytoplasmic inclusion bodies in the dopaminergic nigral neurons, which are called Lewy bodies, and are the pathological hallmark of PD [53]. Levels of total α-synuclein in CSF are significantly decreased [90,91,92,93,94], whereas those of oligomeric and phosphorylated forms are significantly increased in the CSF and plasma of PD patients compared with non-PD patients and control subjects [95,96,97], suggesting that the change of oligomer/total-α-synuclein concentrations may be a useful biomarker for the early diagnosis of PD [96]. Although a significant increase in the levels of NEFL in CSF, plasma, and sera from PD patients can be a potential biomarker distinguishing from control subjects, a large overlap in the values between PD and disease control patients, including PSP and MSA [94,98,99], suggests that an increase of NEFL cannot be a disease biomarker distinguishing PD from other diseases with parkinsonism. [98,100,101]. None of the candidate biomarker proteins, including parkinsonism-associated deglycase (DJ-1), brain-derived neurotrophic factor (BDNF), and glutathione [102,103] have been established as diagnostic biomarkers of PD.

RNAs have also been investigated as possible biomarkers of PD. MiRNAs have been investigated as possible diagnostic biomarker candidates in PD [104]; the upregulation of miR-153 (in CSF), miR-205 (in CSF), miR-30a (in whole blood), miR-26a (in whole blood), miR-16-2 (in whole blood), and miR-29a (in PBMCs) and the downregulation of miR-1 (in CSF and whole blood), miR-29a (in CSF and serum), miR-22 (in whole blood), miR-124 (in serum), miR-221 (in serum), and miR-126 (in PBMCs) [105] have been reported, but with a lack of consistency among the reports.

## 4. Liquid Biopsy in Sporadic ALS

Comprehensive analysis of the ALS patient CSF proteome suggested that changes in extracellular proteins may be candidates for various types of biomarkers [106,107,108,109]. Some of these extracellular proteins, including TDP-43, NEFL, neurofilament heavy chain (NEFH), and extracellular domain of the common neurotrophin receptor p75 (p75^ECD^) are summarized in Table 1.

### 4.1. Diagnostic Biomarkers

TDP-43 pathology or the mislocalization of TDP-43 from the nucleus to the abnormal inclusion in the cytoplasm is the pathological diagnostic hallmark of sporadic ALS, indicating that extracellular molecules that reflect TDP-43 mislocalization would be good diagnostic biomarkers. The TDP-43 concentration in the CSF of ALS patients is significantly higher than that of normal control subjects, Guillain-Barre syndrome, MS patients, and PD patients [110,111,112,113,114], whereas exosomal TDP-43 levels in CSF were not different between the normal control subjects and ALS patients [114].

Neurofilaments, including NEFL and NEFH, have been investigated as possible biomarkers of sporadic ALS [115]. Mutations in the *NEFL* gene cause Charcot-Marie-Tooth disease type 2 (CMT2) and mutations of *NEFH* and neurofilament hyperphosphorylations are associated with ALS, AD, and PD [116]. Several studies indicated that NEFL and phosphorylated NEFH (pNEFH) levels were significantly increased in the CSF and serum from ALS patients compared with normal control subjects and disease control patients [117,118,119,120,121,122,123,124], and the extent of the increase in ALS was highest among neurodegenerative diseases, including AD and PD [119,124], and as high as in chronic inflammatory demyelinating polyneuropathy (CIDP) [119,123,124].

Activation of the neurotrophin receptor p75 in response to the nerve growth factor released by activated astrocytes plays a role in developmental motor neuron death and neuronal injury [125]. The levels of urinary p75^ECD^ are increased in ALS patients compared with normal control subjects and patients with other neurological diseases such as PD and MS [126,127,128]. The levels of neurotrophin receptor p75 in tissues are upregulated also in AD, ischemic stroke, and seizure, suggesting a response to non-specific neuronal injury and cell stress [129].

Therefore, changes in TDP-43, NEFL, pNEFH, and urinary p75^ECD^ levels in body fluids reflect adverse cellular events that are not specifically associated with ALS pathogenesis, suggesting that they may not become disease-specific diagnostic biomarkers.

### 4.2. Other Types of Biomarkers

Changes in TDP-43, NEFL, pNEFH, and urinary p75^ECD^ levels are potential prognostic and/or disease progression biomarkers.

Reduced TDP-43 levels (≤27.9 ng/mL) in CSF are associated with shorter survival times [112], while higher NEFL and pNEFH levels in the CSF and serum are associated with more rapid progression speed and poorer prognosis [117,120,123,124], and higher urinary p75^ECD^ levels are associated with more rapid progression [126,127,128]. Urinary p75^ECD^ levels increase linearly with disease progression from the time of diagnosis and are correlated with clinical disease parameters, ALS functional rating scale-revised (ALSFS-R) [120,127]. Changes in NEFL and pNEFH levels in CSF, but not in serum, from patients with fast and immediate disease progression speed increased with disease duration and ALSFS-R [120,124], suggesting that these changes may be disease progression biomarkers.

These extracellular proteins may become potential biomarkers for prognosis and/or disease progression of sporadic ALS, when proven by additional comprehensive studies in the future.

## 5. RNAs as Biomarker Candidates of Sporadic ALS

Many of the so-far identified ALS-linked genes are involved in RNA processing [28], but only a small proportion of sporadic ALS patients carry any of these mutated genes, including *C9ORF72*, *FUS*, *TDP-43*, and ataxin 2 (*ATXN2*). Abnormal GGGGCC hexanucleotide repeat expansion in the *C9ORF72*, which is the most common gene associated with familial and sporadic ALS in Western countries, alters the RNA binding activity of the translated C9ORF72 protein through the formation of RNA foci [130,131]. Mutations in the *SOD-1* gene, the common genetic cause among familial ALS patients and identified in 2–7% of sporadic ALS patients, affect the expression levels of vascular endothelial growth factor-A (VEGFA) mRNA and NEFL mRNA [15,28]. FUS and TDP-43 are involved in the transcription and splicing of RNAs, and mutations in the encoding genes are identified in a few percentage points of familial ALS and in approximately 1% of sporadic ALS [15]. Additionally, other genes involved in RNA regulation, including *ATXN2* and matrin 3 (*MATR3*), are found to be causative genes of familial ALS, and are also identified in sporadic ALS [131]. Therefore, RNA dysregulation mechanisms are the plausible pathogenesis of sporadic ALS, as well as familial ALS. As the role of extracellular RNAs in the cell–cell communication has recently been highlighted, extracellular RNAs are hopeful new biomarker candidates of sporadic ALS (Table 2 and Table 3).

### 5.1. Diagnostic Biomarkers

Comprehensive studies on the expression levels of mRNA in whole blood and PBMCs reported that the expression levels of as many as 2943 mRNAs [132] and 2300 mRNAs in whole blood [133], and those of 87 mRNAs in PBMCs were significantly different among ALS patients, the normal control subjects, and the disease control patients [134]. However, the different RNAs were not the same among these reports.

Expression levels of the below-mentioned 13 mRNAs that are potentially involved in the pathogenesis of ALS, have been compared between ALS and normal control subjects and disease control patients by using quantitative polymerase chain reaction (qPCR) (Table 2) [135,136,137,138].

VEGFA was suggested to play a role in neuroprotection via activating VEGF receptor 2 and neuropilin-1 in various neurodegenerative diseases, including ALS [139] and mice with a homozygous deletion in the hypoxia response element of the VEGFA promoter exhibited classical ALS-like symptoms [140]. C-C motif Chemokine ligand 2 (CCL2) was also suggested to play a role in neuroprotection from excitotoxicity, by reducing glutamate release from the presynapses and/or facilitating the re-uptake of glutamate in the astrocytes at synapses [141]. Kinesin family member 5C (KIF5C) and dynactin subunit 1 (DCTN1) are motor proteins of axonal flow, and their defects have been shown to be associated with motor neuron death [142,143]. Studies using qPCR [135,136], but not those using RNA sequencing (RNAseq), reported a significant increase in the expression levels of all these four mRNAs (VEGFA mRNA, CCL2 mRNA, KIF5C mRNA, and DCTN1 mRNA) in the PBMCs from ALS patients.

Neurotrophic factors, including BDNF, neurotrophic receptor tyrosine kinase 2 (NTRK2), phosphatidylinositol-4,5-bisphosphate 3-kinase catalytic subunit alpha (PIK3CA), AKT serine/threonine kinase 1 (AKT1), glycogen synthase kinase 3β (GSK3β), nuclear factor κB (NFκB), and Fas ligand (FASLG), play roles in the development and synaptic plasticity of the nervous system, suggesting pathogenic roles in neurological diseases [144]. Indeed, studies using qPCR demonstrated an increase of the expression levels of mRNAs of all these neurotrophic factors in peripheral blood leukocytes (PBL) and four of them (BDNF mRNA, PIK3CA mRNA, AKT1 mRNA, and NFκB mRNA) in whole blood from ALS patients [138].

Cytoplasmic fragile X mental retardation interacting protein 2 (CYFIP2) and retinoblastoma binding protein 9 (RBBP9) are apoptosis regulatory proteins. CYFIP2 evokes p53-induced apoptosis [145], and RBBP9 competes with RB binding to E2F transcription factor 1 (E2F-1) and evokes E2F-1-induced apoptosis [146]. Expression levels of CYFIP2, mRNA, and RBBP9 mRNA were significantly increased in PBL and whole blood from ALS patients [137].

All of the above-mentioned proteins have been proposed to play important roles in other neurodegenerative diseases as well [136,147,148,149], suggesting that the changes in expression levels of these 13 mRNAs would reflect relatively non-selective pathological processes in the CNS. Due to the lack of disease specificity in ALS, these mRNAs cannot be disease-specific diagnostic biomarkers.

Similar to mRNA, miRNAs play important roles in motor neuron homeostasis [150], and hence have been investigated as potential diagnostic biomarker candidates of sporadic ALS [27]. Due to their stability in the extracellular space, miRNAs are considered to be better biomarker candidates than mRNAs. Although a number of studies have reported the differences in the expression levels of miRNAs between control and ALS patients in body fluids [27], only a few of them reported differences in the expression levels of miRNAs that were associated with motor neuron homeostasis (Table 3) [151,152,153,154,155,156,157,158,159,160].

MiR-9 plays an important role in the regulation of axonal development via suppressing microtubule-associated protein 1B (MAP1B) [161], in the differentiation of spinal motor neurons via repressing the transcription factors onecut1 (OC1) and forkhead box P1 (FoxP1) [162,163], in cytoskeletal integrity via regulating the expression of NEFH mRNA [164], and in neuron regeneration via regulating the expression of monocyte chemotactic protein-1-induced protein-1 (MCPIP1) [165]. Studies have shown that the expression levels of miR-9 were significantly increased in the CSF, plasma, and PBL from ALS patients [151,152,153].

MiR-124 mediates the development of spinal motor neurons by silencing repressor element-1 silencing transcription factor (REST) [166], and the regeneration of neurons by upregulating crucial genes. MiR-124 also serves as an indirect regulator of expression of excitatory amino acid transporter 2 (EAAT2) mRNA [167]. The expression levels of miR-124 have been shown to be significantly increased in the CSF of ALS patients [152].

MiR-146a is a critical regulator of expression of NEFL mRNA that is involved in the maintenance of cytoskeletons [168]. Studies have shown that the expression levels of miR-146a were significantly decreased in serum, but had inconsistency in the CSF of ALS patients [152,154,155].

MiR-128 reduces Ras homolog family member A (RhoA) activity by silencing Rho guanine nucleotide exchange factor (PDZ-RhoGEF), leading to neurite growth promotion [169]. The expression level of miR-128 is significantly decreased in the whole blood from ALS patients [156].

MiR-183 is a key regulator of neurite growth by suppressing the expression of mammalian target of rapamycin (mTOR) mRNA [170], and its expression levels have been shown to be significantly decreased in whole blood and PBL from ALS patients [156,157].

MiR-206 is a skeletal muscle-specific miRNA that plays a role in the maintenance and regeneration of the neuromuscular junction [171]. Also, the expression levels of miR-206 were significantly increased in serum and PBL from ALS patients [151,153,158,159].

MiR-338-3p is involved in the apoptosis of mature neuron and the neurodegeneration of oligodendrocytes due to modulating apoptosis-associated tyrosine kinase (AATK) mRNA levels in neurons. In addition, miR-338-3p suppresses the expression levels of SLC1A2 mRNA, suggesting a role in glutamate clearance [172]. Compared with control subjects, the expression levels of miR-338-3p were significantly increased in the serum, PBL, and CSF of ALS patients, and in PBL from other neurodegenerative disease such as AD and PD, as well [172,173].

MiR-133b plays an important role in the differentiation and degeneration of midbrain dopaminergic neurons and promotes neuritis outgrowth [174,175]. The expression levels of miR-133b were significantly increased in the serum and plasma of ALS patients [176].

As similar changes have been detected in the body fluids from patients with neurodegenerative diseases other than ALS, and because some studies demonstrated no change in the levels of these miRNAs [1], it is not likely that the changes in these six miRNA in body fluids from ALS patents are disease-specific. Although all of them are associated with motor neuron homeostasis, their roles in the pathogenetic mechanism of sporadic ALS are largely unknown.

### 5.2. Prognostic Biomarkers

High expression levels of COL19A1 mRNA in whole blood are associated with fast progression of the disease, especially sporadic ALS [177]. Therefore, COL19A1 mRNA may be a prognostic biomarker candidate. Higher expression levels of serum miR-206 are associated with a slower worsening of the Medical Research Council Score, which reflects muscles strength in ALS patients [178]; meanwhile, the expression levels of miR-9 in skeletal muscles are increased in the ALS patients with slow disease progression [179]. Conversely, the expression levels of miR-133b are negatively correlated with the vital capacity [176]. As miRNAs—including miR-9 and miR-146a—regulate the expression levels of NEFL [168], which is a prognostic biomarker candidate, these two miRNAs may be promising, but as yet have not established prognostic biomarker candidates.

### 5.3. Disease Progression Biomarkers

The expression levels of VEGFA, mRNA, and CCL2 mRNA in ALS patients with respiratory dysfunction or definite ALS patients based on El Escorial criteria are higher than those without respiratory dysfunction or probable/possible ALS patients [135], suggesting that the changes in the expression levels of VEGFA, mRNA, and CCL2 mRNA may be disease progression biomarkers of sporadic ALS. Although the expression levels of miR-206 in skeletal muscles are inversely correlated with the time from the symptom onset to muscle biopsy [179], the expression levels of miR-206 in serum do not significantly change with time [159].

Long non-coding RNAs (lncRNAs) are the transcripts of non-coding gene regions greater than 200 nucleotides in length. LncRNAs interact with RNAs, including mRNA, miRNA, and circRNA, and hence are suggested to play roles in ALS pathogenesis [180,181]. There is a report exploring the expression levels of lncRNAs extracted from PBMCs, indicating that 293 lncRNAs were significantly different in the expression levels between normal control subjects and sporadic ALS patients; 183 lncRNAs were upregulated, and the remaining 110 lncRNAs were downregulated [134].

Although many RNAs are proposed as biomarker candidates of sporadic ALS, none of them has been deemed reliable. Most of these studies merely compared the expression levels of molecules among normal control subjects, disease control patients, and ALS patients without mentioning the potential roles in the pathogenesis. Although studies based on disease-specific molecular mechanisms are required for the establishment of biomarkers of ALS, a lack of knowledge about molecular cascades leading motor neurons to death in ALS has limited the targeting of promising molecules. Moreover, the difficulty of detecting small amounts of RNAs and proteins that are released from motor neurons, which are progressively decreasing in number, has prevented the extensive scrutiny of biomarkers.

## 6. Biomarker Candidates Based on Excitotoxicity in Sporadic ALS

Excitotoxicity resulting from the dysregulation of glutamatergic signaling has long been proposed as one of the pathogenic mechanisms of both sporadic and familial ALS [182,183], historically because glutamate levels were increased in post mortem tissue [184] and CSF of ALS patients [185,186,187]. The upper motor neurons use glutamate as neurotransmitters and glutamate secreted from the presynaptic boutons in the axon terminal activates the glutamate receptors, such as the α-amino-3-hydroxy-5-methyl-4-isoxazole propionic acid (AMPA) receptors and the N-methyl-D-aspartate (NMDA) receptors in the dendritic spines of the lower motor neurons, thereby transmitting excitatory signals. The excitatory signals are terminated by the uptake of released glutamate into surrounding astrocytes and/or neurons through glutamate transporters, including EAAT2 (Figure 2) [182]. Therefore, increased extracellular glutamate or the altered function of glutamate receptors may lead to an excessive influx of extracellular ions, including Ca^2+^, into the lower motor neurons. Since an exaggerated increase of intracellular Ca^2+^ concentration is toxic to neurons, the prolonged over-excitation of glutamate receptors may ultimately lead to motor neuron death [183]. Therefore, an increase of glutamate secretion, a reduction of glutamate reuptake, or exaggerated ion influx through aberrant glutamate receptors may be a potential cause of excitotoxicity and cause the death of motor neurons. These lines of evidence suggest that changes in extracellular proteins and RNAs reflecting excitotoxic events would be potential biomarker candidates of sporadic ALS (Figure 3 and Table 4).

The change of extracellular glutamate levels has been reported. The levels of glutamate in the serum, plasma, and CSF of ALS patients have been shown to be significantly increased compared with those from normal control subjects and patients with Hirayama disease [188,189,190]. However, the levels of glutamate in CSF were increased only in approximately 40% of the patients [191], and high levels of glutamate in serum, plasma, and CSF were detected in ischemic stroke and MS patients [192,193] as well, suggesting that extracellular glutamate levels would not be disease-specific diagnostic biomarkers of sporadic ALS. Moreover, the levels of glutamate in plasma are not correlated with disease severity [188] or riluzole treatment [190], suggesting that the levels of glutamate may not serve as prognostic, predictive, or pharmacodynamic biomarkers.

Regarding the glutamate transporter dysfunction, decreased high-affinity glutamate transport [194], a decrease in expression levels of EAAT2 [195], the expression of abnormal intron 7-including EAAT2 mRNA [196], and an increase of EAAT2 pre-mRNA with RNA edited intron 7 [197] have been reported in ALS patients (Figure 2). Moreover, the inhibition of glutamate re-uptake with glutamate transporter inhibitors has been shown to be toxic to rat cortical and spinal motor neurons [198,199]. Mice lacking *glt1*, the rodent ortholog of human EAAT2, exhibited spontaneous seizure, the degeneration of hippocampal neurons, and early death [200]. However, several lines of evidence failed to demonstrate glutamate transporter dysfunction as a primary pathogenesis of ALS; intrathecal administration of glutamate transporter inhibitors causes no significant neuronal loss [201,202]. Furthermore, decreased expression levels of EAAT2 are not specific to ALS; they are also observed in other neurological diseases, including AD, ischemic hypoxia, and traumatic brain injury [203]. Aberrant RNA splicing of EAAT2 was also observed in non-neurological diseases [204]. Indeed, although the expression levels of miR-124 that regulate the expression levels of EAAT2 mRNA were significantly increased in the CSF of ALS patients [152], glutamate uptake in platelets were decreased not only in ALS patients, but also in AD and PD patients [188,205,206], and the EAAT2 levels in platelets from ALS patients were not statistically significantly different compared with those from control patients [207].

Regarding the altered function of glutamate receptors, exaggerated Ca^2+^ influx through the AMPA receptor and/or the NMDA receptor has been proposed to play a role in the death of motor neurons [208]. Ca^2+^ influx through AMPA receptors caused slow motor neuron death, and it has been reported that motor neurons were more vulnerable to AMPA receptor-mediated injury than spinal dorsal horn neurons in the primary neuronal culture, and the administration of AMPA receptor antagonists suppressed motor neuron death due to various causes [199,209]. Meanwhile, Ca^2+^ influx through the NMDA receptors is involved in rapid cell death such as epilepsy and encephalitis, and the administration of NMDA receptor agonists fails to cause neuronal loss [198,210], suggesting that AMPA receptors may play a more significant role than NMDA receptors in slow motor neuron death, as seen in ALS patients (Figure 2) [210].

AMPA receptors are comprised of homotetramers or heterotetramers of GluA1, GluA2, GluA3, and GluA4. Ca^2+^ permeability of the AMPA receptor is determined by the presence of GluA2 with its glutamine/arginine (Q/R) site edited in the subunit assembly [211]. The CAG codon at the Q/R site of the GluA2 transcript is converted to CIG by the activity of adenosine deaminase acting on RNA 2 (ADAR2), that specifically convert adenosine in the Q/R site of GluA2 pre-mRNA into inosine [212]. This post-transcriptional A-to-I conversion is called RNA editing, which occurs ubiquitously but most actively in the mammalian CNS, including in humans [213]. Since inosine is recognized as guanosine during translation, A-to-I conversion at the Q/R site of GluA2 mRNA results in the change from the glutamine codon (CAG) to that for arginine (CGG). Arginine (R) instead of glutamine (Q) at the Q/R site of GluA2 protein alters the ion-permeability property of the AMPA receptors, changing from Ca^2+^-permeable to Ca^2+^-impermeable [214,215,216]. As this conversion normally occurs at the Q/R site of all the GluA2 expressed in the motor neurons, AMPA receptors expressed in motor neurons always include Q/R site-edited GluA2 and AMPA receptors expressed in motor neurons, which are Ca^2+^-impermeable [214,215,216]. However, GluA2 mRNA that is expressed in the spinal motor neurons of sporadic ALS patients are not always edited at the Q/R site because of the downregulation of ADAR2 [217,218]. The downregulation of ADAR2 is specific, and the expression levels of ADAR1 mRNA—another member of the ADAR family—in the motor neurons are not different between ALS and normal control subjects [218]. ADAR2 downregulation is specific in the motor neurons of sporadic ALS patients, and is not observed in other neurons of ALS patients or the motor neurons of normal control subjects and disease control patients, including patients with spinal and bulbar muscle atrophy (SBMA) and those with MSA, and transgenic rats with mutated human SOD1 [217,218,219]. Moreover, TDP-43 pathology, the pathological hallmark of sporadic ALS, is always associated with the motor neurons lacking immunoreactivity to ADAR2 in patients with sporadic ALS [220]. Furthermore, the roles of calpain in the mechanisms whereby TDP-43 mislocalization occurs in the ADAR2-lacking motor neurons were demonstrated in the conditional ADAR2 knockout mice [221]. The disease specificity and the link to TDP-43 pathology render ADAR2 downregulation a plausible hypothesis of the ALS pathogenesis.

It has been reported that RNAs with ADAR2-dependent sites in extracellular RNAs reflect intracellular ADAR2 activity in vitro [222], indicating that the change in ADAR2-dependent RNA editing of extracellular RNAs is a promising diagnostic biomarker of sporadic ALS.

ADAR2 downregulation with expression of Q/R site-unedited GluA2 mRNA was reported in the motor neurons of familial ALS patients carrying the *FUS*^P525L^ mutation as in sporadic ALS [223]. In addition, exaggerated Ca^2+^ influx through AMPA receptors [224] and ADAR2 mislocalization from the nucleus to the cytoplasm, with inactive A-to-I conversion, have been reported to be associated with enhanced hexanucleotide repeat expansion in *C9ORF72* [225]. Therefore, biomarker candidates based on excitotoxicity would be biomarkers of some familial ALS. Moreover, therapy based on the correction of fatal molecular mechanisms induced by the downregulation of ADAR2 has been developed, including the adeno-associated virus vector-based ADAR2 gene therapy [226] and the inhibition of increased Ca^2+^ influx through abnormal AMPA receptors with the AMPA receptor antagonist perampanel [227], and we will know the results of clinical trials of perampanel by early 2020. Therefore, the changes in editing efficiencies at the ADAR2-dependent A-to-I sites in extracellular RNAs may represent predictive, pharmacodynamic, and diagnostic biomarkers of sporadic ALS.

## 7. Conclusion and Future Directions

In this review, we summarized current knowledge regarding the role of extracellular proteins and RNAs as candidate biomarkers of neurodegenerative diseases, especially ALS. We currently have no reliable biomarkers of sporadic ALS, although the levels of Aβ42, T-tau, and P-tau in AD and those of α-synuclein in PD have been very promising biomarkers based on pathogenesis. As there is growing evidence that RNA dysregulation is involved in the pathogenesis of ALS, changes in the expression levels and processing of extracellular RNAs, reflecting excitotoxicity, would be very promising diagnostic, predictive, and/or pharmacodynamic biomarkers.

## Figures and Tables

**Figure 1 ijms-20-03148-f001:**
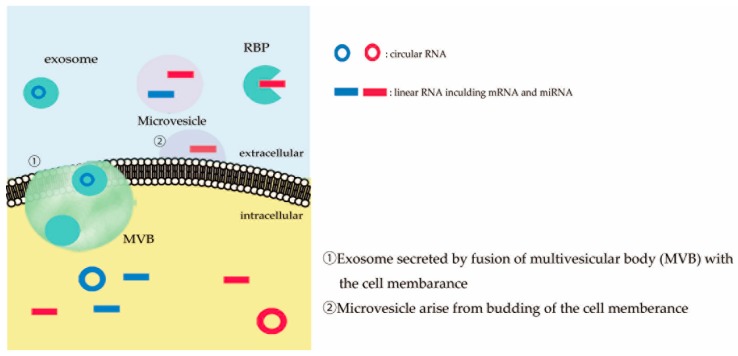
Extracellular RNAs. RNAs are found extracellularly, as extracellular vesicles or RNA binding proteins (RBP)–RNA complexes. Circles and lines indicate circular RNAs and linear RNAs, respectively.

**Figure 2 ijms-20-03148-f002:**
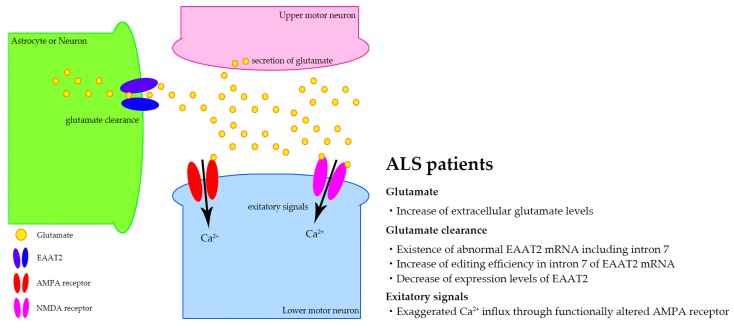
Excitotoxic mechanisms proposed in sporadic amyotrophic lateral sclerosis (ALS). In the synaptic cleft, glutamate is released from the axon terminal of the upper motor neuron (pink), and binds to the glutamate receptors such as the α-amino-3-hydroxy-5-methyl-4-isoxazole propionic acid (AMPA) receptor and the N-methyl-D-aspartate (NMDA) receptor expressed in the dendritic spines of the lower motor neuron (blue). The excitatory signals are regulated by glutamate transporters, including excitatory amino acid transporter 2 (EAAT2), which is expressed in astrocytes or neurons (green). An increase in glutamate levels in the postmortem tissue and cerebrospinal fluids, a decrease of glutamate clearance through EAAT2 due to aberrant mRNA processing, including intron 7 inclusion and an increase of editing efficiency in intron 7, and a decrease of expression levels of EAAT mRNA are shown. Also, motor neuron death due to exaggerated Ca^2+^ influx through functionally altered AMPA receptors is shown.

**Figure 3 ijms-20-03148-f003:**
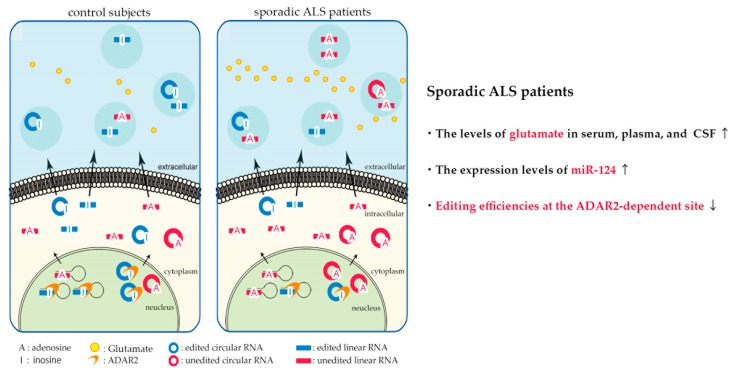
Biomarker candidates based on excitotoxicity in amyotrophic lateral sclerosis (ALS). A decrease of the RNA editing activity at the adenosine deaminase acting on RNA 2 (ADAR2)-dependent sites in the extracellular RNAs would be diagnostic, predictive, and pharmacodynamic biomarkers of sporadic ALS. This figure is modified from a previously published figure [223].

**Table 1 ijms-20-03148-t001:** Proteins considered as biomarker candidates of sporadic amyotrophic lateral sclerosis (ALS). ALSFS-R: ALS functional rating scale-revised, CSF: cerebrospinal fluid, TDP-43: transactive response DNA/RNA binding protein of 43kDa, FTLD: frontotemporal lobar degeneration.

Proteins	Changes in Levels	Kinds of Body Fluids	Biomarkers	Patients (Origin and Number)		Disease Specificity	Reference
TDP-43	Increase	CSF	Diagnostic	Germany	15 sporadic ALS	Low (not disease-specific)	[110]
					12 FTLD		
					9 ALS + FTLD		
					13 disease controls		
				Japanese	30 sporadic ALS		[111]
					29 disease controls		
				Japanese	27 sporadic ALS		[112]
					50 disease controls		
				Japanese	13 sporadic ALS		[113]
					7 GBS		
				Germany	9 sporadic ALS		[114]
					4 FTLD		
					8 control subjects		
			Prognostic	Japanese	29 disease controls	High (correlation with survival time and disease duration)	[111]
					27 sporadic ALS		
				Japanese	27 sporadic ALS		[112]
					50 disease controls		
				Japanese	13 sporadic ALS		[113]
					7 GBS		
Neurofilament (NEFL or pNEFH)	Increase	CSF	Diagnostic	Germany	67 sporadic ALS	Low (not disease-specific)	[117]
					2 familial ALS		
					33 control subjects		
				American	20 sporadic ALS		[118]
					20 control subjects		
				Germany	222 sporadic ALS		[119]
					20 familial ALS		
					199 disease controls		
				Germany	194 sporadic ALS		[120]
					26 familial ALS		
					316 disease controls		
				European	176 sporadic ALS		[121]
					63 disease controls		
				Chinese	53 sporadic ALS		[122]
					32 disease controls		
				American	134 sporadic ALS		[123]
					15 familial ALS		
					101 disease controls		
				Germany	124 sporadic ALS		[124]
					109 disease controls		
					50 control subjects		
			Prognostic	Germany	67 sporadic ALS	High (correlation with survival time and disease duration)	[117]
					2 familial ALS		
					33 control subjects		
				Germany	222 sporadic ALS		[119]
					20 familial ALS		
					199 disease controls		
				Germany	194 sporadic ALS		[120]
					26 familial ALS		
					316 disease controls		
				European	176 sporadic ALS		[121]
					63 disease controls		
				Chinese	53 sporadic ALS		[122]
					32 disease controls		
			Disease progression	Germany	194 sporadic ALS	High (correlation with ALSFS-R)	[120,124]
					26 familial ALS		
					316 disease controls		
				Germany	124 sporadic ALS		[124]
					109 disease controls		
					50 control subjects		
		Serum	Diagnostic	Germany	67 sporadic ALS	Low (not disease-specific)	[117,118,119,120,121,122,123,124]
					2 familial ALS		
					33 control subjects		
				American	20 sporadic ALS		[118]
					20 control subjects		
				Germany	222 sporadic ALS		[119]
					20 familial ALS		
					199 disease controls		
				Germany	194 sporadic ALS		[120]
					26 familial ALS		
					316 disease controls		
				European	176 sporadic ALS		[121]
					63 disease controls		
				Chinese	53 sporadic ALS		[122]
					32 disease controls		
				American	134 sporadic ALS		[123]
					15 familial ALS		
					101 disease controls		
				Germany	124 sporadic ALS		[124]
					109 disease controls		
					50 control subjects		
			Prognostic	European	176 sporadic ALS	High (correlation with survival time and disease duration)	[121]
					63 disease controls		
				American	134 sporadic ALS		[123]
					15 familial ALS		
					101 disease controls		
				Germany	124 sporadic ALS		[124]
					109 disease controls		
					50 control subjects		
p75^ECD^	Increase	Urine	Diagnostic	Australian	28 sporadic ALS	Low (not disease-specific)	[126]
					19 disease controls		
					12 control subjects		
				Australian	54 sporadic ALS		[127]
					45 control subjects		
				Chinese	101 sporadic ALS		[128]
					108 disease controls		
					97 control subjects		
			Prognostic	Australian	28 sporadic ALS	High (correlation with survival time)	[126,127,128]
					19 disease controls		
					12 control subjects		
				Australian	54 sporadic ALS		[127]
					45 control subjects		
				Chinese	101 sporadic ALS		[128]
					108 disease controls		
					97 control subjects		
			Disease progression	Australian	54 sporadic ALS	High (correlation with ALSFS-R)	[127]
					45 control subjects		

**Table 2 ijms-20-03148-t002:** mRNAs related to the pathogenesis of ALS and considered as biomarker candidates of sporadic ALS. PBMCs: plasma and peripheral blood mononuclear cells.

mRNA	Changes in Levels	Kinds of Body Fluids	Patients (Origin and Number)		Disease Specificity	Reference
**Diagnostic biomarkers**						
VEGFA mRNA	Increase	PBMCs	Indian	50 sporadic ALS	Low (not disease-specific)	[135]
				50 control subjects		
CCL2 mRNA	Increase	PBMCs	Indian	50 sporadic ALS	Low (not disease-specific)	[135]
				50 control subjects		
KIF5C mRNA	Decrease	PBMCs	Polish	74 sporadic ALS	Low (not disease-specific)	[136]
				28 disease controls		
				65 control subjects		
DCTN1 mRNA	Increase	PBMCs	Polish	74 sporadic ALS	Low (not disease-specific)	[136]
				28 disease controls		
				65 control subjects		
BDNF mRNA	Decrease	PBL	Dutch	50 sporadic ALS	Low (inconsistent results, not disease-specific)	[132]
				50 disease control		
				50 control subjects		
	Increase	Whole blood	Indian	64 sporadic ALS		[138]
				122 disease controls		
				120 control subjects		
NTRK2 mRNA	Decrease	PBL	Indian	64 sporadic ALS	Low (not disease-specific)	[138]
				122 disease controls		
				120 control subjects		
PIK3CA mRNA	Decrease	PBL	Dutch	50 sporadic ALS	Low (inconsistent results, not disease-specific)	[132]
				50 disease control		
				50 control subjects		
	Increase	Whole blood	Indian	64 sporadic ALS		[138]
				122 disease controls		
				120 control subjects		
AKT1 mRNA	Decrease	PBL	Dutch	50 sporadic ALS	Low (inconsistent results, not disease-specific)	[132]
				50 disease control		
				50 control subjects		
	Increase	Whole blood	Indian	64 sporadic ALS		[138]
				122 disease controls		
				120 control subjects		
GSK3β mRNA	Decrease	PBL	Indian	64 sporadic ALS	Low (not disease-specific)	[138]
				122 disease controls		
				120 control subjects		
NFκB mRNA	Decrease	PBL	American	123 sporadic ALS	Low (inconsistent results, not disease-specific)	[133]
				123 control subjects		
	Increase	Whole blood	Indian	64 sporadic ALS		[138]
				122 disease controls		
				120 control subjects		
FASLG mRNA	Increase	PBL	Indian	64 sporadic ALS	Low (not disease-specific)	[138]
				122 disease controls		
				120 control subjects		
CYFIP2 mRNA	Increase	PBL	Dutch	50 sporadic ALS	Possible (More studies are needed)	[132]
				50 disease control		
				50 control subjects		
		Whole blood	Israeli	6 sporadic ALS		[137]
				3 non-ALS		
RBBP9 mRNA	Increase	PBL	Israeli	6 sporadic ALS	Possible (More studies are needed)	[137]
				3 non-ALS		
**Prognostic biomarker**						
COL19A1 mRNA	Increase	Whole blood	Spanish	59 sporadic ALS	Possible (More studies are needed)	[177]
				24 disease controls		
				58 controls subjects		
**Disease progression biomarkers**						
VEGFA mRNA	Increase	PBMCs	Indian	50 sporadic ALS	High (correlation with respiratory dysfunction)	[135]
				50 control subjects		
CCL2 mRNA	Increase	PBMCs	Indian	50 sporadic ALS	High (correlation with respiratory dysfunction)	[135]
				50 control subjects		

**Table 3 ijms-20-03148-t003:** miRNAs associated with motor neuron homeostasis and considered as biomarker candidates of sporadic ALS. AD: Alzheimer disease, PBL: peripheral blood leukocytes.

miRNA	Changes in Levels	Kinds of Body Fluids	Patients (Origin and Number)		Disease Specificity	Reference
**Diagnostic biomarkers**						
miR-9	Increase	Plasma	American	50 sporadic ALS	Low (not disease-specific)	[151]
				50 AD		
				50 PD		
				50 FTLD		
				50 control subjects		
		CSF	British	32 sporadic ALS		[152]
				6 MS		
				10 control subjects		
		PBL	Slovenian	77 sporadic ALS		[153]
				7 familial ALS		
				27 control subjects		
miR-124	Increase	CSF	British	32 sporadic ALS	Low (not disease-specific)	[152]
				6 MS		
				10 control subjects		
miR-146a	Decrease	CSF	British	32 sporadic ALS	Low (inconsistent results, not disease-specific)	[152]
				6 MS		
				10 control subjects		
	Increase	PBMCs	American	22 sporadic ALS		[154]
				4 familial ALS		
				24 control subjects		
	Decrease	Serum	Italian	14 sporadic ALS		[155]
				8 control subjects		
miR-128	Decrease	Whole blood	Italian	50 sporadic ALS	Low (not disease-specific)	[156]
				15 control subjects		
miR-183	Decrease	Whole blood	Italian	50 sporadic ALS	Low (not disease-specific)	[156]
				15 control subjects		
		PBL	Chinese	83 sporadic ALS		[157]
				24 PD		
				61 control subjects		
miR-206	Increase	plasma	American	50 sporadic ALS	Low (not disease-specific)	[151]
				50 AD		
				50 PD		
				50 FTLD		
				50 control subjects		
		PBL	Slovenian	77 sporadic ALS		[153]
				7 familial ALS		
				27 control subjects		
		Serum	Spanish	12 sporadic ALS		[158]
				12 control subjects		
		Serum	British	27 sporadic ALS		[159]
				36 disease controls		
				25 control subjects		
miR-338-3p	Increase	Serum	Italian	72 sporadic ALS	Low (not disease-specific)	[172]
		PBL		62 control subjects		
		CSF				
		PBL	Italian	14 sporadic ALS		[173]
				14 control subjects		
miR-133b	Increase	Serum	American	20 sporadic ALS	Low (not disease-specific)	[176]
				3 familial ALS		
				30 control subjects		
**Prognostic biomarkers**						
miR-206	Increase	Plasma	Brazilian	39 sporadic ALS	High (correlation with Medical Research Council Score)	[178]
				39 control subjects		
miR-9	Increase	Plasma	American	50 sporadic ALS	Possible (More studies are needed)	[151]
				50 AD		
				50 PD		
				50 FTLD		
				50 control subjects		
		CSF	British	32 sporadic ALS		[152]
				6 MS		
				10 control subjects		
		PBL	Slovenian	77 sporadic ALS		[153]
				7 familial ALS		
				27 control subjects		
miR-133b	Increase	Serum	American	20 sporadic ALS	Possible (More studies are needed)	[176]
				3 familial ALS		
				30 control subjects		
**Pharmacodynamic biomarkers**						
miR-9	Increase	Plasma	American	50 sporadic ALS	Possible (More studies are needed)	[151]
				50 AD		
				50 PD		
				50 FTLD		
				50 control subjects		
		CSF	British	32 sporadic ALS		[152]
				6 MS		
				10 control subjects		
		PBL	Slovenian	77 sporadic ALS		[153]
				7 familial ALS		
				27 control subjects		
miR-206	Increase	Plasma	American	50 sporadic ALS	Possible (More studies are needed)	[151]
				50 AD		
				50 PD		
				50 FTLD		
				50 control subjects		
		PBL	Slovenian	77 sporadic ALS		[153]
				7 familial ALS		
				27 control subjects		
		Serum	Spanish	12 sporadic ALS		[158]
				12 control subjects		
		Serum	British	27 sporadic ALS		[159]
				36 disease controls		
				25 control subjects		
**Disease progression biomarker**						
miR-206	Increase	Serum	British	36 disease controls	Low (no correlation with disease progression)	[159]
				25 control subjects		

**Table 4 ijms-20-03148-t004:** Biomarker candidates based on excitotoxicity.

	Changes in Levels	Kinds of Body Fluids	Reliability	Reference
**Diagnostic biomarkers**				
Glutamate	Increase	Serum	Low (not disease-specific)	[188,189,190,191]
		Plasma		
		CSF		
miR-124	Increase	CSF	Low (not disease-specific)	[152]
Editing efficiencies at the ADAR2-dependent sites	Decrease	In vitro	Possible (More studies are needed)	[223]
**Prognostic biomarkers**				
Glutamate	Increase	Plasma	Low	[188]
**Predictive biomarkers**				
Glutamate	Increase	Plasma	Low	[191]
Editing efficiencies at the ADAR2-dependent sites	Decrease	In vitro	Possible (More studies are needed)	[223]
**Pharmacodynamic biomarkers**				
Glutamate	Increase	Plasma	Low	[191]
Editing efficiencies at the ADAR2-dependent sites	Decrease	In vitro	Possible (More studies are needed)	[223]

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
