# Peer review of "Extracellular RNAs as Biomarkers of Sporadic Amyotrophic Lateral Sclerosis and Other Neurodegenerative Diseases"

_ijms, 2019, doi:10.3390/ijms20133148_

Round 1
Reviewer 1 Report
The paper is of interest.
Specific Comments
The title of the paper is misleading . The authors do not provide an overview of biomarkers but rather concentrate on mRNA markers etc. Either they need to change the title to better reflect the context of their review or to discuss other markers such as inflammation, oxidative stress, genetics, epigenetic etc.
Additional tables discussing major findings of original papers, their main findings, cohorts tested, origin of patients etc will help the reader to better understand the work performed in this field.
Author Response
Dear Assistant editor and Reviewer1
We are grateful to have heard your favorable decision with appropriate comments of the reviewers. We carefully revised our manuscript, tables (Table1, Table2, and Table3), and figures (Figure1, Figure2, and Figure3) in line with reviewers’ comments as described below.
All the changes are colored in blue and underlined in the text.
The paper is of interest.
Specific Comments
The title of the paper is misleading . The authors do not provide an overview of biomarkers but rather concentrate on mRNA markers etc. Either they need to change the title to better reflect the context of their review or to discuss other markers such as inflammation, oxidative stress, genetics, epigenetic etc.
Additional tables discussing major findings of original papers, their main findings, cohorts tested, origin of patients etc will help the reader to better understand the work performed in this field.
We thank the reviewer for valuable comment. We changed the title of the paper “Extracellular RNAs as biomarkers of sporadic amyotrophic lateral sclerosis and other neurodegenerative diseases”.
Additionally, we added some information in the tables to help the readers‘ better understanding.
Reviewer 2 Report
The manuscript presented by Hosaka and coworkers shows a revision of molecular biomarkers related to different neurodegenerative diseases, such as Alzheimer (AD), Parkinson (PD) and Amyotrophic Lateral Sclerosis (ALS) disesases. In particular, this manuscript focuses on biomarkers (extracellular proteins, mRNA, microRNA and circular RNA) related to sporadic ALS.
The manuscript is well organized and the information provided by the authors is clearly exposed. However, some comments need to be addressed:
- The title of the manuscript suggest that the information provided in it is mainly focused on sporadic ALS but the authors widely explain in the first part of the manuscript the state of the art in PD and AE. In addition, a small section about Extracellular RNAs (section 2) explains the importance of different RNAs in cell homeostasis. Therefore, the tittle should better reflect the general content of the manuscript.
- In section 3, subsection 3.1 (Alzheimer Disease), in the last paragraph the authors could explain more in detail the main RNAs candidates for biomarkers of AD as it appears in the next section about PD. There is extensive bibliography about non-coding RNAs, especially miRNAs, related to AD that can be of help to better explain this subsection.
- In section 5 and in particular in subsections 5.2 and 5.3, a recent published article shows that COL19A1 mRNA is related to the disease progression in ALS patients and especially in sporadic ALS patients, suggesting this biomarker as a potential prognostic biomarker in ALS. This study could be also included in this section (Calvo AC, et al, 2019, doi: 10.14336/AD.2018.0917).
- In section 6, it is not clear if the biomarkers described are related to both familial and sporadic ALS cases or only sporadic ALS cases. Please, revise this point. In addition, lines 421-422 should be rephrased.
- An additional section discussing the presence of common biomarkers in both familial and sporadic ALS can be of help to understand the role and nature of these biomarkers (extracellular proteins and RNAs) in both forms of the disease.
Author Response
Dear Assistant Editor and Reviewer2
We are grateful to have heard your favorable decision with appropriate comments of the reviewers. We carefully revised our manuscript, tables (Table1, Table2, and Table3), and figures (Figure1, Figure2, and Figure3) in line with reviewers’ comments as described below.
All the changes are colored in blue and underlined in the text.
The title of the manuscript suggest that the information provided in it is mainly focused on sporadic ALS but the authors widely explain in the first part of the manuscript the state of the art in PD and AE. In addition, a small section about Extracellular RNAs (section 2) explains the importance of different RNAs in cell homeostasis. Therefore, the tittle should better reflect the general content of the manuscript.
We thank the reviewer for valuable comment. We changed the title of the paper as “Extracellular RNAs as biomarkers of sporadic amyotrophic lateral sclerosis and other neurodegenerative diseases”.
- In section 3, subsection 3.1 (Alzheimer Disease), in the last paragraph the authors could explain more in detail the main RNAs candidates for biomarkers of AD as it appears in the next section about PD. There is extensive bibliography about non-coding RNAs, especially miRNAs, related to AD that can be of help to better explain this subsection.
We thank the reviewer for valuable comment. We modified the sentences in the last paragraph in section 3.1 (Page 5, line 200-212 in the revised manuscripts) as follows and added a reference in order to facilitate readers’ understanding; Ramakrishna, S. et al, 2019
Page 5, line 200-212 in the revised manuscripts
“Multiple lines of study report significant differences in the levels of miRNAs in the CSF or serum between patients with AD and control subjects [83-87]; upregulation of let-7d (in plasma and whole blood), let-7f (in serum), let-7g (plasm), miR-106a (in serum), miR-106b (in serum), miR-142 (in whole blood), miR-15b (in plasma), miR-191 (in plasma), miR-3065 (in serum), miR-34a (in plasma), miR-483 (in plasma), miR-9 (in serum), miR-127 (in serum), and miR-30c (in CSF) and downregulation of let-7d (in serum), let-7g (in serum), miR-106a (in whole blood), miR-106b (in whole blood), miR-107 (in plasma and whole blood), miR-125b (in serum and plasma), miR-146a (in serum and plasma), miR-15b (in serum), miR-181c (in serum), miR-191 (in serum), miR-26a (in serum), miR-29b (in plasma and peripheral blood mononuclear cells (PBMCs)), miR-3065 (in plasma), miR-342 (in serum and plasma), miR-483 (in serum), miR-9 (in whole blood and CSF), miR-127 (in CSF), miR-184 (in CSF), miR-195 (in CSF) have been reported. However, inconsistency among results has failed to establish diagnostic biomarkers of AD.”
- In section 5 and in particular in subsections 5.2 and 5.3, a recent published article shows that COL19A1 mRNA is related to the disease progression in ALS patients and especially in sporadic ALS patients, suggesting this biomarker as a potential prognostic biomarker in ALS. This study could be also included in this section (Calvo AC, et al, 2019, doi: 10.14336/AD.2018.0917).
We thank the reviewer for valuable comment. we added a reference; Calvo AC, et al, 2019 and modified the sentences (Page 11, line 381-383 in the revised manuscripts) as follows.
Page 11, line 381-383 in the revised manuscripts
“High expression levels of COL19A1 mRNA in whole blood are associated with fast progression of the disease, especially sporadic ALS. Therefore, COL19A1 mRNA may be prognostic biomarker candidate.”
- In section 6, it is not clear if the biomarkers described are related to both familial and sporadic ALS cases or only sporadic ALS cases. Please, revise this point. In addition, lines 421-422 should be rephrased.
Although excitotoxicity resulting from dysregulating of glutamatergic signaling has been proposed as one of the pathogenic mechanisms of both sporadic and familial ALS, the biomarker candidates, levels of glutamate in serum and expression levels of miR-124, are related to sporadic ALS. However, ADAR2 down-regulation is related to familial and sporadic ALS. To clear the biomarker candidates described in section 6 are mainly related to sporadic ALS, we modified section 6 title as follows. And we modified the sentences (Page 16, line 442-444 in the revised manuscripts)
“6. Biomarker candidates based on excitotoxicity in sporadic ALS”
Page 16, line 442-444 in the revised manuscripts
“The change of extracellular glutamate levels has been reported. The levels of glutamate in serum, plasma and CSF of ALS patients have been shown to be significantly increased compared with those from normal control subjects and Hirayama disease”
Reviewer 3 Report
Review Manuscript ID: ijms-518048
Hosaka et al. Biomarkers of sporadic amyotrophic lateral sclerosis and other neurodegenerative diseases.
This review describe the importance to discover new biomarkers in the neurodegenerative diseases, essential for diagnosis and therapeutic development. New biomarkers have been listed. However only extracellular proteins and RNA have been considered as biomarkers of sporadic ALS.
The review is well written although there are some points to be better discussed.
For instance at Line 219 it is not clear the reason why NEFL cannot be a disease biomarker.
A general revision is needed to clarify some unclear points.
All the figures (Fig. 1 -2 -3) need an accurate description of the mechanisms and more details.
I think also that other reviews and papers need to be cited.
Several candidate markers were found in either cerebrospinal fluid (CSF) or blood (serum/plasma). Combinations of biomarkers would reflect the status of different tissues, including motor neuron, inflammation, muscle health and metabolism (Vijayakumar et al., A Systematic Review of Suggested Molecular Strata, Biomarkers and Their Tissue Sources in ALS. Front Neurol. 2019 May 14;10:400).
Also for the biomarkers found in muscle tissues, more details should be given. Indeed, it has been found that miRNA206 is increased in skeletal muscle of MLC/SOD mice (a mouse model carrying the SOD1 mutation exclusively in skeletal muscle) confirming muscle specificity (Dobrowolny et al. Muscle Expression of SOD1(G93A) modulates microRNA and mRNA transcription pattern associated with the myelination process in the spinal cord of transgenic mice. Front Cell Neurosci. 2015 Dec 1;9:463). In the same mice model ion channels are proposed as biomarkers of the pathology (Camerino et al. Elucidating the contribution of skeletal muscle ion channels to Amyotrophic Lateral Sclerosis in search of new therapeutic options. Sci Rep. 2019 Feb 28;9(1):3185)
Author Response
Dear Assistant Editor and Reviewers
We are grateful to have heard your favorable decision with appropriate comments of the reviewers. We carefully revised our manuscript, tables (Table1, Table2, and Table3), and figures (Figure1, Figure2, and Figure3) in line with reviewers’ comments as described below.
All the changes are colored in blue and underlined in the text.
This review describe the importance to discover new biomarkers in the neurodegenerative diseases, essential for diagnosis and therapeutic development. New biomarkers have been listed. However only extracellular proteins and RNA have been considered as biomarkers of sporadic ALS.
The review is well written although there are some points to be better discussed.
For instance at Line 219 it is not clear the reason why NEFL cannot be a disease biomarker.
A general revision is needed to clarify some unclear points.
All the figures (Fig. 1 -2 -3) need an accurate description of the mechanisms and more details.
I think also that other reviews and papers need to be cited.
Several candidate markers were found in either cerebrospinal fluid (CSF) or blood (serum/plasma). Combinations of biomarkers would reflect the status of different tissues, including motor neuron, inflammation, muscle health and metabolism (Vijayakumar et al., A Systematic Review of Suggested Molecular Strata, Biomarkers and Their Tissue Sources in ALS. Front Neurol. 2019 May 14;10:400).
Also for the biomarkers found in muscle tissues, more details should be given. Indeed, it has been found that miRNA206 is increased in skeletal muscle of MLC/SOD mice (a mouse model carrying the SOD1 mutation exclusively in skeletal muscle) confirming muscle specificity (Dobrowolny et al. Muscle Expression of SOD1(G93A) modulates microRNA and mRNA transcription pattern associated with the myelination process in the spinal cord of transgenic mice. Front Cell Neurosci. 2015 Dec 1;9:463). In the same mice model ion channels are proposed as biomarkers of the pathology (Camerino et al. Elucidating the contribution of skeletal muscle ion channels to Amyotrophic Lateral Sclerosis in search of new therapeutic options. Sci Rep. 2019 Feb 28;9(1):3185)
We thank the reviewer for valuable comment. Although NEFL may be a biomarker distinguishing patients with parkinsonism and healthy control, NEFL cannot be a biomarker distinguishing PD and other diseases with parkinsonism. To avoid readers’ misunderstanding, we modified the sentences (Page 5, line 227 to Page 6, line 231) as follows.
Page 5, line 227 to Page 6, line 231
“Although a significant increase in the levels of NEFL in CSF, plasma and sera from PD patients can be a potential biomarker distinguishing from control subjects, a large overlap in the values between PD and disease control patients, including PSP and MSA, suggests that an increase of NEFL cannot be a disease biomarker distinguishing PD from other diseases with parkinsonism.”
Additionally, we modified Figure 1, Figure 2, and Figure 3 for better reader’s understanding.
Moreover, we added biomarker candidates, miR-338-3p and miR-133b, and modified the relevant sentences (Page 9, line 366-374 and Page 9, line 387) as follows. Accordingly, we added references; Vijayakumar et al, 2019, De Felice, B et al, 2012, De Felice, B et al, 2014, Lui, X. C et al, 2015, Kim, J et al, 2007, and Raheja, R et al, 2018.
Page 9, line 366-374
“MiR-338-3p is involved in the apoptosis of mature neuron and the neurodegeneration of oligodendrocyte due to modulate apoptosis-associated tyrosine kinase (AATK) mRNA levels in neuron. In addition, miR-338-3p suppresses the expression levels of SLC1A2 mRNA, suggesting a role in glutamate clearance. Compared with control subjects, the expression levels of miR-338-3p were significantly increased in serum, PBL, and CSF from ALS patients, and in PBL from other neurodegenerative disease such as AD and PD, as well.
MiR-133b plays an important role in differentiation and degeneration of midbrain dopaminergic neurons and promotes neuritis outgrowth. The expression levels of miR-133b were significantly increased in serum and plasma from ALS patients.”
Page 9, line 3870
“Conversely, expression levels of miR-133b negatively correlated with the vital capacity.”